# Imperfect gold standard gene sets yield inaccurate evaluation of causal gene identification methods
Lijia Wang, Xiaoquan Wen ⓘ ✉ & Jean Morrison ⓘ ✉

Causal gene discovery methods are often evaluated using reference sets of causal genes, which are treated as gold standards (GS) for the purposes of evaluation. However, evaluation methods typically treat genes not in the GS positive set as known negatives rather than unknowns. This leads to inaccurate estimates of sensitivity, specificity, and AUC. Labeling biases in GS gene sets can also lead to inaccurate ordering of alternative causal gene discovery methods. We argue that the evaluation of causal gene discovery methods should rely on statistical techniques like those used for variant discovery rather than on comparison with GS gene sets.

Identifying causal genes for complex diseases can highlight disease-specific dysregulated pathways, improve disease classification, and identify drug targets[1]. In genetic association analysis, it has become a common practice to implicate putative causal genes (PCG) computationally by linking variant-level genetic association evidence and the existing biological knowledge base[2]. Some methods approach PCG implication as a supervised learning problem, aiming to predict unknown binary causal/non-causal gene labels for a given trait, while others rank the candidate genes by their likelihood of being PCGs, returning a continuous probability estimate or ranking for each gene. In this article, we focus on the common practice of evaluating PCG implication methods in reference to known sets of causal genes. While many papers making use of these sets for evaluation acknowledge that reference sets may be incomplete, this is rarely accounted for in evaluation techniques, where they are treated as gold-standard (GS)[2,3]. A critical challenge for this assessment strategy is that known causal genes may differ meaningfully from as-yet unidentified causal genes.

Table 1 summarizes methods used by recent publications to identify GS-positive genes. Genes assigned as gold standard positives by these methods are often reliable and based on stringent standards for causality. However, when proposed methods are evaluated against these GS gene sets, genes not labeled as positive are implicitly treated as non-causal or negative. These genes are almost certainly contaminated by some as-yet-unidentified causal genes (thus motivating continued PCG discovery research). In fact, the more confident we are in the positive labels of a GS gene set, the more mislabeled non-causal genes we can expect[2]. GS gene sets also tend to favor genes with particular features determined by the method of constructing the set. For example, GS gene sets derived from the set of causal coding variants favor genes that act through protein-coding changes rather than expression regulatory mechanisms[3]. Most classification methods also have gene-feature-related biases due to the type of data they use as input. A PCG implication method will appear more accurate if it is evaluated using a GS

gene set with similar feature-related biases to its own and less accurate if the GS gene set has different biases. Authors naturally select a GS gene set constructed using features they feel are important and may, therefore, unintentionally tilt the scales toward their own proposed method.

We show that when the GS gene set is incomplete, estimates of power, specificity, and receiver operating characteristic (ROC) are inaccurate and may even misorder the relative quality of two different classifiers. This phenomenon can occur even if the GS gene set contains no false positives. We argue that no true GS sets of labeled genes are currently available. Therefore, we urge caution in interpreting comparisons of causal gene classifiers based on existing labels.

## Effect of label contamination on evaluation metrics
### Evaluation with PU-labeled gene sets
Genes outside the constructed GS set are more accurately viewed as unlabeled (U) rather than as negatives (N). Combined with accurately positively labeled genes in the GS set, the overall GS gene set should be regarded as positive-unlabeled (PU) data, a term used in semi-supervised machine learning. Using PU data to evaluate performance as though they were positive-negative (PN) labeled data results in inaccurate evaluations[4]. A PU-labeled gene set with perfect positive labeling consists of three subsets of genes: true causal genes that are correctly identified (labeled positives), true causal genes that are not labeled and therefore assumed to be non-causal (unlabeled positives), and non-causal genes that are unlabeled and therefore correctly assumed to be non-causal (unlabeled negatives, UN) (Fig. 1a). Evaluation treating PU labels with perfect positive labeling as PN labels will always under-estimate the positive predictive value (precision) and overestimate the negative predictive value (NPV) of a classifier (Fig. 1b).

Figure 2 shows four possibilities for the performance of the classifier on the unlabeled positive genes and the corresponding relationship between the

Department of Biostatistics, University of Michigan, Ann Arbor, MI, USA. ✉e-mail: xwen@umich.edu; jvmorr@umich.edu

## Table 1 | Methods used for construction of gold standard gene sets

| Construction scheme | Publication |
| --- | --- |
| Genes harboring large-effect-size coding variants for selected phenotypes derived from OMIM and additional literature. | Connally et al.[12] |
| Genes that are targets of any known drug against the disease of interest (reported in Open Targets). | Picart-Armada et al.[2] |
| Genes annotated to phenotypes of interest in databases (OMIM, GO, DrugBank, literature-derived drug targets) | Greene et. al. (2015)[13] Tranchevent et al.[14] |
| Phenotype-associated genes compiled from GWAS catalog mining, publication-based, and expert curation. | Kolosov, Daly, and Artomov[4] |
| Genes derived from expert domain knowledge, drug target-disease pairs, experimental alteration, and functional data. | Mountjoy et al.[15] |
| Genes with fine-mapped protein-coding variants within loci non-coding credible sets. | Weeks et al.[3] |
| Top 10% genes ranked by PoPS for validation. | Gazal et al.[16] |

All listed methods used their constructed gold standard gene sets to compare the precision and recall of classification methods or evaluate the AUC of ranking methods. Kolosov, Daly, and Artomov[4] makes use of both ROC curves and the PU score.

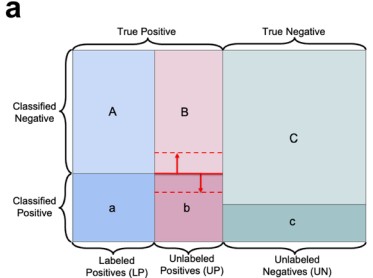

**a**

**b**

|  | True Value | Estimated Value |
| --- | --- | --- |
| Sensitivity | $\frac{a+b}{a+b+A+B}$ | $\frac{a}{A+a}$ |
| Specificity | $\frac{C}{C+c}$ | $\frac{B+C}{b+c+B+C}$ |
| PPV | $\frac{a+b}{a+b+c}$ | $\frac{a}{a+b+c}$ |
| NPV | $\frac{C}{A+B+C}$ | $\frac{B+C}{A+B+C}$ |

**Fig. 1 | Illustration of classifier performance relative to an imperfectly labeled gene set. a** Illustration of gene classes by reference and classifier labels. The rectangle represents the set of total genes, with columns corresponding to the three types of labeled genes: correctly labeled positive genes (blue, A/a), positive genes labeled as negatives (red, B/b), and correctly labeled negative genes (green C/c). Horizontal divisions indicate classifications, with the lower divisions (lowercase letters, darker shading) classified as positive and the upper divisions (capital letters, lighter shading) classified as negative. The classifier may not have the same performance on unlabeled positive genes as it has on labeled positive genes (red lines and arrows), which affects estimated sensitivity and specificity. **b** Comparison of true and estimated sensitivity, specificity, positive predictive value (PPV), and negative predictive value (NPV) when some positives are unlabeled.

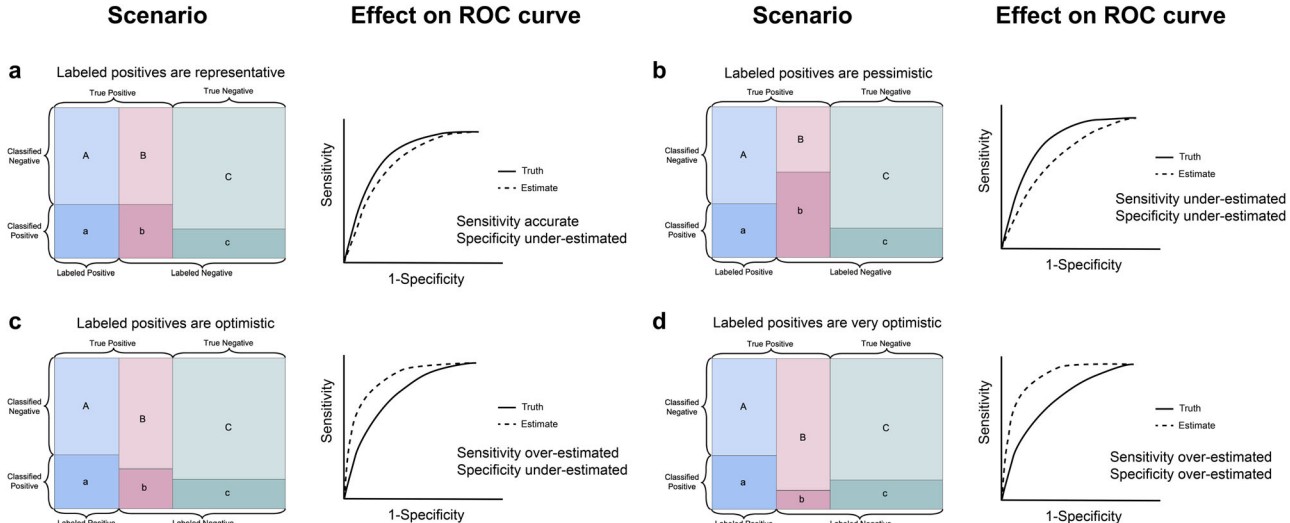

**Fig. 2 | Effect of PU labeling on estimated ROC curves.** The rectangle in each figure represents the set of total genes, as in Fig. 1. The drawing on the right of each figure represents the expected relationship between the true and estimated ROC curve for each scenario. **a** Labeled positives are representative of all true positive genes. **b** The classifier is more sensitive to unlabeled positives than labeled positives. **c** The classifier is more sensitive to labeled positives than unlabeled positives. **d** The classifier detects a lower proportion of unlabeled positives than true negatives. Note that in scenario (**c**), the estimated ROC curve could be either above, below, or intersecting with the true ROC curve.

estimated and true sensitivity, specificity, and ROC curve. The classifier may perform differently on unlabeled positive genes and labeled positive genes if these two groups differ on important features that either align with or do not align with the features used to construct the classifier.

In almost all scenarios, estimation with PU labels leads to under-estimating specificity. To see why this is, let A, B, C, a, b, and c be defined as illustrated in Fig. 1 as the counts of labeled positive, unlabeled positive, and true negative genes that are classified as non-causal (upper-case) or causal

## a

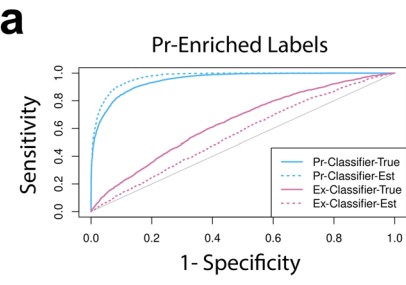

Pr-Enriched Labels

## b

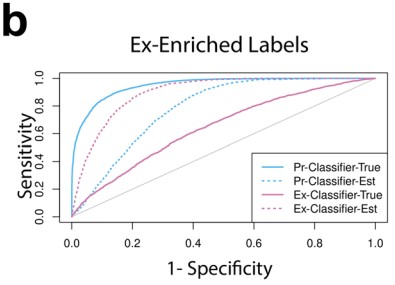

Ex-Enriched Labels

**Fig. 3 | Comparison of Classifiers on True vs. Biased Labels.** Classifiers evaluated against true labels (solid lines) and biased labels (dotted lines). In both figures, the Pr-Classifier uses Pr values to identify causal genes and the Ex-Classifier uses Ex values. **a** Causal genes with large Pr values are more likely to be labeled as positive than causal genes with large Ex values. Performance of the Pr-Classifier is overestimated while performance of the Ex-classifier is underestimated. **b** Causal genes with large Ex values are more likely to be labeled as positive than causal genes with large Pr values. Performance of the Ex-Classifier is overestimated, making it appear more accurate than the Pr-classifier.

(lower-case) by the classifier. With these definitions, $\alpha = \frac{B+b}{C+c}$ is the proportion of unlabeled genes that are truly causal. As long as the sensitivity of the classifier to unlabeled positives is higher than the probability of falsely predicting a true negative to be causal, $B/(B+b) < C/(C+c)$, so $B < \alpha C$. Therefore,

$$\text{Spec}_{True} = \frac{C}{C+c} = \frac{(1+\alpha)C}{(1+\alpha)(C+c)} \tag{1}$$

$$> \frac{B+C}{(1+\alpha)(C+c)} = \frac{B+C}{B+b+C+c} = \text{Spec}_{Estimated} \tag{2}$$

This means that specificity is underestimated in all cases except for the scenario in Fig. 2d. However, sensitivity may be either over- or underestimated depending on the feature biases of the labeled positive genes. If the classifier is more sensitive to unlabeled positives than to labeled positives (Fig. 2b), the sensitivity will be underestimated. If the classifier is less sensitive to unlabeled positives than to labeled positives (Fig. 2c, d), the sensitivity will be overestimated.

Errors in estimating sensitivity and specificity result in errors in the ROC curve and, therefore, in the area under the ROC curve (AUC). These errors also affect other measures that rely on the 2 × 2 confusion matrix, such as Matthew's correlation coefficient and F1 score. This error applies to evaluating ranking methods as well as methods that return only hard classifications. In the special case in Fig. 2a, the classifier has an equal ability to detect labeled and unlabeled positive genes, so sensitivity is estimated accurately. Motivated by this observation, refs. 4,5 rely on a "PU score" which is analogous to the F1 score but reliant only on sensitivity and not on specificity. However, if labeled positive genes are not representative of all positive genes, the PU score will also be inaccurate.

In genetics research, we expect labeling biases because there are multiple molecular mechanisms by which a causal gene can affect complex diseases, and different classification and GS identification methods will favor different mechanisms. For example, as shown in Table 1, several GS gene set construction strategies focus on genes with phenotype-associated coding variants. Genes that affect phenotypes primarily through expression dysregulation may not be represented in these GS gene sets, so classifiers particularly sensitive to causal genes acting through expression changes may appear to perform poorly when using these gene sets.

### Simulated example
To illustrate this issue, we consider a hypothetical example in which each gene has two continuous, measurable features, *Pr* and *Ex*. We think of these features as continuous summaries of the evidence that a gene acts on the trait through mechanisms mediated by either protein sequence (*Pr*) or expression level (*Ex*). Let $Y_i$ be a binary indicator that gene $i$ is causal for the trait of interest. We simulate $Pr_i$ and $Ex_i$ from independent standard normal distributions and generate $Y_i$ as

$$Y_i \sim Bern(\pi_i)$$
$$logit(\pi_i) = -3 + 6Pr_i + 2Ex_i + \epsilon_i.$$

In our simulation, the protein feature, *Pr* is a stronger predictor of causality than the expression feature, *Ex*.

In each simulated data set, we generate *Pr*, *Ex*, and causal status, *Y* for 20,000 genes that are divided into a set of 10,000 genes used for training and 10,000 genes used for testing. In the training set, we fit two classifiers, the Pr-classifier, and the Ex-classifier, by fitting a logistic regression with *Y* as the outcome and either *Pr* or *Ex* only as a predictor. This differs from the methods generally used to build causal gene discovery methods, as no perfectly labeled gene sets are available for training. However, this strategy provides a straightforward method to obtain classifiers based on only one of the two gene features.

The 10,000 genes in the testing set function as our GS gene set. We consider 3 possibilities. Either all genes are correctly labeled, positives with high levels of *Pr* are more likely to be correctly labeled, or positives with high levels of *Ex* are more likely to be correctly labeled. We refer to these as correct, Pr-enriched, and Ex-enriched labels. Let $Z_{C,i}$, $Z_{Pr,i}$, and $Z_{Ex,i}$ be correct, Pr-enriched, and Ex-enriched labels for gene $i$ in the testing set. We generate these as $Z_{C,i} = Y_i$, $Z_{Pr,i} = Y_i W_{Pr,i}$, and $Z_{Ex,i} = Y_i W_{Ex,i}$ with

$$W_{Pr,i} \sim Bern(\theta_{Pr,i}) \qquad W_{Ex,i} \sim Bern(\theta_{Ex,i})$$
$$logit(\theta_{Pr,i}) = -3 + 4P_i \quad logit(\theta_{Ex,i}) = -3 + 4E_i.$$

The Pr-enriched labels mislabel 4.5% of all positives as negative, while the Ex-enriched labels mislabel 13.5% of all positives as negatives.

ROC curves estimated using each of the imperfect label sets are shown in Fig. 3, compared against ROC curves estimated using perfect labels. In both cases, label enrichment results in biased estimation of classifier performance. When Pr-enriched labels are used, the AUC of the Pr-classifier is overestimated, and the AUC of the Ex-classifier is underestimated. However, the accuracy of the two classifiers is correctly ordered. When the Ex-enriched labels are used, the pattern is reversed, resulting in the misordering of the two classifiers. These results align with our theoretical expectations. When using the Pr-enriched labels to evaluate the Pr-classifier or the Ex-enriched labels to evaluate the Ex-classifier, we have the scenario in Fig. 2c, where sensitivity is over-estimated, and specificity is under-estimated, pushing the ROC curve up from its true value. Conversely, when label enrichment does not favor the genes a classifier is most sensitive to, we are in the scenario in Fig. 2b, where sensitivity is under-estimated, pushing the ROC curve down.

### Outlook
It is currently impossible to confidently determine a comprehensive GS gene set that includes all causal genes for any trait due to the myriad biological mechanisms leading to complex phenotypes. Several studies have

acknowledged that supervised ML methods designed to classify PCGs should not be trained on non-comprehensive GS gene sets[4,6,7]. Here, we draw attention to the fact that sensitivity and specificity estimated using non-comprehensive GS gene sets are also inaccurate, making it inappropriate to compare and evaluate methods using these or related measures such as AUC and F1 scores. However, properly evaluating PCG implication methods is critical to making progress in this field. To address a similar issue in other fields, researchers have proposed incorporating negative controls (i.e., known negatives) or weights estimating each unit's probability of being detected based on its features[8,9]. These methods do not clearly extend to the PCG implication problem. It may be possible to build a case for some negative control gene-trait pairs. However, these will likely differ meaningfully from unknown negatives, leading to similar issues of biased estimation. Using weighting in this context would require estimating the probability that the gene was labeled given its features for each labeled positive gene, which is impossible without knowledge of the feature distribution of all true positive genes.

An alternative approach circumventing the issue is to use a statistical model-based approach for causal gene identification. This is a common approach in the field of causal variant identification, where methods for statistical fine-mapping rely on probabilistic models, allowing them to obtain model-based measures of uncertainty, such as posterior inclusion probabilities or confidence intervals[10,11]. Probabilistic methods can also be evaluated in simulations to test their robustness to violations of modeling assumptions. Using probabilistic models validated in simulations is the most practical method for obtaining defensible estimates of a method's false discovery rate under different parameter settings.

Incomplete GS gene sets may still play an important role in method evaluation, as they can be used to provide an empirical estimate of the sensitivity of a method at a given parameter setting corresponding to a known false discovery rate or to compute the PU score used by ref. 4. However, it is important to note that this is a context-specific measure of sensitivity that may not replicate in other gene sets with different features. When presenting sensitivity results, researchers should acknowledge potential feature biases of GS gene sets and evaluate their methods using multiple gene sets constructed from different information.

Finally, incomplete GS gene sets should not be used to construct ROC curves or compute measures such as the F1 statistic. This limitation of PU-labeled data provides an argument against the use of purely rank-based PCG implication methods. Methods that supply only a ranking and no model-based measure of false discovery rate are completely reliant on accurately labeled testing data for calibration. We have argued that no such testing data exists, meaning it is impossible to calibrate rank-only methods to a target false discovery rate. Instead, we urge researchers to develop PCG implication methods to make use of statistical approaches that provide model-based measures of label uncertainty.

## Code availability

Code replicating simulations can be found in the Supplementary Data and at https://lijiaw.gitlab.io/GS-gene-sets/comparison.html.

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

## Author contributions
L.W.: Literature review, conducting simulations, manuscript preparation, X.W.: Advising, manuscript preparation, J.M.: Advising, manuscript preparation, figure creation.

## Competing interests
The authors declare no competing interests.
