## [Peer Review File · Communications Biology]

Reviewers' comments:

Please note that Referees #1-2 both have expertise in statistical genetics and genomics

Reviewer #1 (Remarks to the Author):

In this paper, Wang et al noticed that labeling biases in gold-standard gene sets can lead to inaccurate ordering of discovery methods. The authors argue that evaluation of these methods need to consider the incompleteness of the GS sets.

The message is important, and the results are worth publishing. I just have a few questions.

In Abstract, the authors claim that "We argue that evaluation of these methods should rely on statistical techniques like those used for variant discovery". I wonder what statistical techniques the authors are referring to? Some elaboration on this point would be appreciated.

The authors only look at ROC, which is incomplete. I suggest the authors also look at PRC and other metrics such as Matthew's Correlation coefficient (MCC) and F1 score.

Is it possible to show some real data examples instead of just simulation? This will help the readers to put this important message in context.

Reviewer #2 (Remarks to the Author):

The current manuscript sheds light on an important topic of biases and inaccuracies arising due to imperfection of "gold-standard" genes. Through a small set of simulation and intuitive explanations the authors demonstrate their point about inaccurate sensitivity and specificity in using treating unlabeled genes as "true negatives". The report, as it is written, is extremely clear in its approach and direct in making the point.

Minor points: (A) I would not call the existing genes to be "gold-standard". Most works using such gene sets acknowledge that there is an element of noise present in selecting these genes as well as a potential issue of power in their initial identification. Is it possible to use any other nomenclature for this "gold-standard" or cite an example publication where this has been named "gold-standard"?

(B) Since the authors have used "P" for positive in the initial paragraphs I would encourage them to not use P for Protein classifier.

(C) Beyond the AUC curves in Appendix it might be good to look at the difference in sensitivity/specificity as a function of the P/E enrichment.

An entirely optional comment: For some cases the gold standard causal gene is known. For example if the outcome is protein levels, we can safely assume that the underlying gene is causal with highly likelihood. Can the authors demonstrate their point on such real-world setting to see whether inaccuracies in sensitivity/specificity is as big a problem as manifested in the simulations?

We'd like to thank the editor and reviewers for their time evaluating this paper. Our responses are below in blue.

Reviewer #1 (Remarks to the Author):

In this paper, Wang et al noticed that labeling biases in gold-standard gene sets can lead to inaccurate ordering of discovery methods. The authors argue that evaluation of these methods need to consider the incompleteness of the GS sets.

The message is important, and the results are worth publishing. I just have a few questions.

In Abstract, the authors claim that “We argue that evaluation of these methods should rely on statistical techniques like those used for variant discovery”. I wonder what statistical techniques the authors are referring to? Some elaboration on this point would be appreciated.

Thank you for this suggestion. We have added a clarification in the main body of text: “An alternative approach that circumvents the issue is to use a statistical model-based approach for causal gene identification.

This is a common approach in the field of causal variant identification, where methods for statistical fine-mapping rely on probabilistic models, allowing them to obtain model-based measures of uncertainty, such as posterior inclusion probabilities or confidence intervals. Probabilistic methods can also be evaluated in simulations to test their sensitivity to violations of modeling assumptions.”

The authors only look at ROC, which is incomplete. I suggest the authors also look at PRC and other metrics such as Matthew's Correlation coefficient (MCC) and F1 score.

Thank you for these suggestions. While we did not include a precision-recall curve for the simulated example, but we do discuss positive predictive value (precision) and sensitivity (recall), which is also shown in the ROC curve. We show that PPV will always be under-estimated using PU labels compared with PN labels. We have added sentence indicating that our results extend beyond the ROC curve and AUC to any measure of accuracy that make use of the 2x2 confusion matrix:

“Error in estimating sensitivity and specificity results in error in the ROC curve and therefore error in the area under the ROC curve (AUC), as well as error in other measures which rely on the 2×2 confusion matrix such as Matthew's correlation coefficient and F1 score.”

Is it possible to show some real data examples instead of just simulation? This will help the readers to put this important message in context.

Thank you for this suggestion. This paper was motivated by several instances of recent papers (Table 1) performing analyses in which genes not previously identified as causal were treated as gold-standard negatives. The aim of this paper was to point out the potential pitfalls of doing so.

The primary constraint in this problem is lack of availability of truly gold-standard negative gene sets. As a result, it is not possible to do a side-by-side comparison of analysis using Positive/Unlabeled genes and Positive/Negative genes in real data.

Reviewer #2 (Remarks to the Author):

The current manuscript sheds light on an important topic of biases and inaccuracies arising due to imperfection of “gold-standard” genes. Through a small set of simulation and intuitive explanations the authors demonstrate their point about inaccurate sensitivity and specificity in using treating unlabeled genes as “true negatives”. The report, as it is written, is extremely clear in its approach and direct in making the point.

Minor points: (A) I would not call the existing genes to be “gold-standard”. Most works using such gene sets acknowledge that there is an element of noise present in selecting these genes as well as a potential issue of power in their initial identification. Is it possible to use any other nomenclature for this “gold-standard” or cite an example publication where this has been named “gold-standard”?

Thank you, this is a very good point. We have retained the term gold-standard but have clarified our usage in two places. In the abstract:

“Causal gene discovery methods are often evaluated using reference sets of causal genes, which are treated as gold-standards (GS) for the purposes of evaluation. However, in practice, these gene sets are always incomplete, leading to mis-estimation of sensitivity, specificity, and AUC.”

And in the main text:

“In this article, we focus on the common practice of evaluating PCG implication methods in reference to known sets of causal genes. While many papers making use of these sets for evaluation acknowledge that reference sets may be incomplete, this is rarely accounted for in evaluation techniques, where they are treated as gold-standard (GS).

A critical challenge for this assessment strategy is that known causal genes may differ meaningfully from as-yet unidentified causal genes.”

(B) Since the authors have used “P” for positive in the initial paragraphs I would encourage them to not use P for Protein classifier.

Thank you for pointing this out. We have changed the abbreviation for protein classifier and expression classifier to Pr-Classifier and Ex-Classifier respectively.

(C) Beyond the AUC curves in Appendix it might be good to look at the difference in sensitivity/specificity as a function of the P/E enrichment.

Thanks. We only looked at four combinations of settings presented, with the intent of demonstrating the range of possibilities. However, we added a sentence to the supplement discussing the relationship between label enrichment and the accuracy of estimated model performance:

“These observations align with the insights from Figure A1, which also allow us to infer how the relationship between true and estimated sensitivity changes as classifier enrichment changes. Regardless of the true feature effect, Pr-enriched labels always lead to overestimation of the performance of the Pr-classifier and under-estimation of the performance of the Ex-Classifer, while the opposite is true for Ex-enriched labels. This is because, when labeling bias and classifier bias align, we are in the scenario in either Figure A1a or b, where sensitivity is under-estimated. If labeling bias and classifier bias are misaligned, we will be in scenario A1c, where sensitivity is underestimated.

An entirely optional comment: For some cases the gold standard causal gene is known. For example if the outcome is protein levels, we can safely assume that the underlying gene is causal with high likelihood. Can the authors demonstrate their point on such real-world setting to see whether inaccuracies in sensitivity/specificity is as big a problem as manifested in the simulations?

Thanks - this is a good suggestion. However, the point would be best demonstrated in a situation in which all the causal genes are known. Even in cases where we have very confident knowledge of some causal genes, it may be hard to rule out that there are additional unidentified causal genes. For example, in the case of protein levels, the underlying gene is almost certainly causal, but there could be additional other causal genes that regulate protein levels through transcriptional regulation or through regulating the degradation and metabolism process. We are not aware of any traits for which there is a confident complete list of causal genes. However, we are interested in continuing this investigation in future work.

REVIEWERS' COMMENTS:

Reviewer #1 (Remarks to the Author):

The responses from the authors seem adequate.

Reviewer #2 (Remarks to the Author):

The authors have addressed my comments satisfactorily. I commend them for an important and insightful study.